# Learning Multi-resolution Functional Maps with Spectral Attention for Robust Shape Matching

**Lei Li**
LIX, École Polytechnique, IP Paris
`lli@lix.polytechnique.fr`

**Nicolas Donati**
LIX, École Polytechnique, IP Paris
`nicolas.donati@polytechnique.edu`

**Maks Ovsjanikov**
LIX, École Polytechnique, IP Paris
`maks@lix.polytechnique.fr`

## Abstract

In this work, we present a novel non-rigid shape matching framework based on multi-resolution functional maps with spectral attention. Existing functional map learning methods all rely on the critical choice of the spectral resolution hyperparameter, which can severely affect the overall accuracy or lead to overfitting, if not chosen carefully. In this paper, we show that spectral resolution tuning can be alleviated by introducing spectral attention. Our framework is applicable in both supervised and unsupervised settings, and we show that it is possible to train the network so that it can *adapt the spectral resolution*, depending on the given shape input. More specifically, we propose to compute multi-resolution functional maps that characterize correspondence across a range of spectral resolutions, and introduce a spectral attention network that helps to combine this representation into a single coherent final correspondence. Our approach is not only accurate with near-isometric input, for which a high spectral resolution is typically preferred, but also robust and able to produce reasonable matching even in the presence of significant non-isometric distortion, which poses great challenges to existing methods. We demonstrate the superior performance of our approach through experiments on a suite of challenging near-isometric and non-isometric shape matching benchmarks.

## 1  Introduction

Shape matching is a critical task in 3D shape analysis and has been paramount to a broad spectrum of downstream applications, including registration, deformation, and texture transfer [1, 2], to name a few. The algorithmic challenge of robust shape matching primarily lies in the fact that shapes may undergo significant variations, such as arbitrary non-rigid deformations. Earlier works to tackle non-rigid shape correspondence conventionally build upon hand-crafted features and pipelines [3], while with the advent of deep learning, the research focus has largely shifted to data-driven and learning-based approaches for improved matching robustness and accuracy [4].

To learn for non-rigid shape matching, a growing body of literature [7, 8, 9, 10, 11, 12] advocates the use of spectral techniques, in particular, the functional map representation [13], which compactly encodes correspondences as small-sized matrices using a reduced spectral basis. A number of advances have been made to the functional map-based networks in terms of probe feature learning [5, 14], differentiable map regularization [5], supervised [7] and unsupervised learning [9, 8, 15], among many others. Despite this progress, existing works nearly always learn functional maps in a single spectral resolution (the number of basis functions used), which is often set empirically. However, the functional map resolution plays a crucial role in the non-rigid shape matching performance,

36th Conference on Neural Information Processing Systems (NeurIPS 2022).

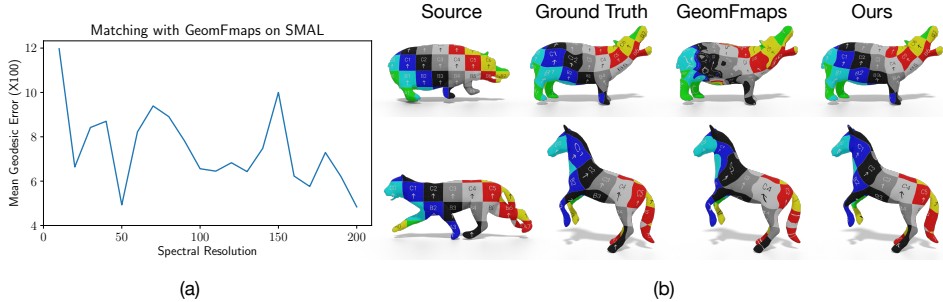

Figure 1: (a) The unstable matching performance of GeomFmaps [5] w.r.t. the critical *Spectral Resolution* hyperparameter on an animal shape dataset SMAL [6]. (b) Correspondence visualization by texture transfer for near-isometric (top) and non-isometric (bottom) shapes. GeomFmaps is trained with ground truth supervision, while our approach is not.

as observed in existing literature [9]. As a concrete example, Fig. 1-(a) shows that the matching performance of a state-of-the-art supervised learning method GeomFmaps [5] fluctuates significantly when trained with a different map resolution (i.e., size of the functional map). Therefore, to improve robustness, it is desirable to enable networks to adaptively change the resolution in a data-dependent manner: for near-isometric shapes, adopting a higher spectral resolution allows high-frequency details to be leveraged for more precise matching; while for non-isometric shapes, adopting a lower spectral resolution is advantageous to obtain approximate but potentially more robust maps.

Motivated by the above discussion, in this work, we propose a novel learning-based functional map framework that learns to adaptively combine multi-resolution maps with a mechanism that we call *spectral attention*, which can accommodate both near-isometric and non-isometric shapes at the same time. Specifically, our framework consists of two novel components (Fig. 2): (1) multi-resolution functional maps and (2) the spectral attention module.

Given as input a pair of non-rigid shapes, we first use a functional map network to estimate a series of maps with varying spectral resolution. Next, we feed the obtained functional maps to a spectral attention network to predict a weight for each map. The attention weights are used to combine all the intermediate maps into a final coherent map. To enable such an assembly, we design a *differentiable spectral upsampling* module that can transform the intermediate maps to the same spectral resolution within a learnable network. Finally, to train our network, we propose to impose penalties on the intermediate multi-resolution functional maps as well as the final map. This is different from existing approaches, e.g., [7, 9, 5, 10, 16], which work with and penalize a single hand-picked spectral resolution. Our method can be trained in both supervised and unsupervised settings and can directly benefit from other advances in deep functional map training, such as improved architectures or regularization. To evaluate our model, we perform a comprehensive set of experiments on several challenging non-rigid shape matching datasets, where our model achieves superior matching performance over existing methods.

In a nutshell, the main contributions of our work are as follows: (1) We introduce a powerful non-rigid shape matching framework equipped with multi-resolution functional maps with spectral attention for handling diverse shape inputs. (2) We propose a novel spectral attention network and a differentiable spectral upsampling module for robust functional map learning. (3) We demonstrate the superior performance of our model compared to existing approaches through extensive experiments on challenging non-rigid shape matching benchmarks. Our code and data are publicly available[1].

## 2   Related Work

In shape analysis, the field of non-rigid shape matching is both extensive and well-studied. In the following paragraphs, we review the works that are most closely related to our approach. A more complete overview can be found in recent surveys [17, 18], and more recently [4] (Section 4).

---

[1] https://github.com/craigleili/AttentiveFMaps

**Functional Maps** Our method is based on the functional maps framework, first introduced in [13], and extended in various works such as [19, 20, 21, 22, 23, 24] among others (see [25] for an overview). This general approach is based on encoded maps between shapes using a reduced basis representation. Consequently, the problem of map optimization becomes both linear and more compact. Besides, this framework allows to represent natural constraints such as near-isometry or bijectivity as linear-algebraic regularization. It has also been extended to the partial setting [26, 27].

One of the bottlenecks of this framework is the estimation of so-called "descriptor functions" that are key to the functional map computation. Early methods have relied on axiomatic features, mainly based on multi-scale diffusion-based descriptors, *e.g.*, HKS and WKS [28, 29].

**Learning-based methods** Several approaches have proposed to learn maps between shapes by formulating it as a dense segmentation problem, e.g., [30, 31, 32, 33, 34, 35]. However, these methods (1) usually require many labeled training shapes, which can be hard to obtain, and (2) tend to overfit to the training connectivity, making the methods unstable to triangulation change.

Closer to our approach are deep shape matching methods that also rely on the functional map framework, pioneered by FMNet [7]. In this work, SHOT descriptors [36] are given as input to the network, whose goal is to refine these descriptors in order to yield a functional map as close to the ground-truth as possible. The key advantage of this approach is that it directly estimates and optimizes for the map itself, thus injecting more structure in the learning problem. FMNet introduced the idea of learning for shape *pairs*, using the same feature extractor (in their case, a SHOT MLP-based refiner) for the source and target shapes in a Siamese fashion to produce improved output descriptors for functional map estimation. However, later experiments conducted in [5] have highlighted that SHOT-based pipelines suffer greatly from connectivity overfitting. Thus, in more recent works, the authors in [5, 15, 14] advocate for learning directly from shapes' geometry, while exploiting strong regularizers for functional map estimation.

The major upside of using the functional map framework for deep shape matching is that it relies on the intrinsic information of shapes, which results in overall good generalization from training to testing, especially across pose changes, which involve minimal intrinsic deformation.

**Unsupervised spectral learning** The methods described above are supervised deep shape matching pipelines. While these methods usually give good correspondence prediction, they need ground-truth supervision at training time. Consequently, other methods have focused on training for shape matching using the functional map framework, *without ground-truth supervision*. This was originally performed directly on top of FMNet by enforcing either geodesic distance preservation [8, 37], or natural properties on the output functional map [9], as well as by promoting cycle consistency [38].

To disambiguate symmetries present in many organic shapes, some works choose to rely on so-called "weak-supervision", by rigidly aligning all shapes (on the same three axes) as a pre-processing step [15, 39], and then use the extrinsic embedding information to resolve the symmetry ambiguity. This, however, limits their utility to correspondences between shapes with the same rigid alignment as the training set. Another solution is to use input signals that are independent to the shape alignment, such as SHOT [36] descriptors as done in the original FMNet. One of these recent methods [11], makes use of optimal transport on top of this SHOT-refiner to align the shapes at different spectral scales. This method, like ours, computes the functional map at different scales via progressive upsampling, but they only keep the last map as the output whereas we propose to let the network learn the best combination of different resolutions. Additionally, this method is dependent on the SHOT input, which makes it unstable towards change in triangulation. In-network refinement is also performed in DG2N [40], but not in the spectral space.

**Attention-based spectral learning** The attention mechanism was originally introduced in deep learning for natural language processing, and consists in putting relative weights on different words of an input sentence [41]. This mechanism can be applied in different contexts, including that of shape analysis. Indeed, attention learned in the feature domain can be used to focus on different parts of a 3D shape, for instance in partial shape matching, as done in [10]. As we show in this paper, attention can also be used in *the spectral domain* by letting the network focus on different levels of details depending on the input shapes and their resulting functional maps at different spectral resolutions. Indeed, the utility of considering different resolutions of a functional map, *e.g.*, via upsampling of its size, has been highlighted in [42, 43]. Here we propose to let the network learn to adaptively combine all the intermediate functional maps into a final coherent correspondence.

## 3   Background

Our work proposes a learning-based framework for non-rigid shape matching by building upon the functional map representation [13], and especially its learning-based variant GeomFmaps, introduced in [5]. Before describing our approach in Sec. 4, we first briefly describe the basic learning pipeline with functional maps for shape correspondence. We refer the interested reader to relevant works [25, 7, 8, 9, 5] for more technical details.

**Deep Functional Map Pipeline**   We consider a pair of shapes $\mathcal{S}_1$ and $\mathcal{S}_2$, represented as triangle meshes with $n_1$ and $n_2$ vertices, respectively. The goal is to compute a high quality dense correspondence between these shapes in an efficient way. The basic learning pipeline estimates a functional map between $\mathcal{S}_1$ and $\mathcal{S}_2$ using the following four steps [25].

- Compute the first $k$ eigenfunctions of the Laplace-Beltrami operator [44] on each shape, which will be used as a basis for decomposing smooth functions on these shapes. The Laplacian is discretized as $S^{-1}W$, where $S$ is the diagonal matrix of lumped area elements for mesh vertices and $W$ is the classical cotangent weight matrix [45]. The eigenfunctions are stored as columns in matrices $\Phi_1 \in \mathbb{R}^{n_1 \times k}$ and $\Phi_2 \in \mathbb{R}^{n_2 \times k}$.

- Second, a set of descriptors (also known as probe, or feature functions) on each shape are extracted by a feature extractor network [7, 5] here denoted by $\mathcal{F}_\Theta$ with learnable parameters $\Theta$. These feature functions are expected to be approximately preserved by the unknown map. We denote the learned feature functions as $\mathcal{F}_\Theta(\mathcal{S}_1) = G_1 \in \mathbb{R}^{n_1 \times d}$ and $\mathcal{F}_\Theta(\mathcal{S}_2) = G_2 \in \mathbb{R}^{n_2 \times d}$, where $d$ is the number of descriptors. After projecting them onto the respective eigenbases, the resulting coefficients are stored as columns of matrices $\mathbf{A}_1, \mathbf{A}_2 \in \mathbb{R}^{k \times d}$, respectively.

- Next, we compute the optimal functional map $\mathbf{C} \in \mathbb{R}^{k \times k}$ by solving:

$$\mathbf{C} = \arg\min_{\mathbf{C}} \|\mathbf{C}\mathbf{A}_1 - \mathbf{A}_2\|^2 + \alpha\|\mathbf{C}\boldsymbol{\Delta}_1 - \boldsymbol{\Delta}_2\mathbf{C}\|^2, \tag{1}$$

  where the first term promotes preservation of the probe functions, and the second term regularizes the map by measuring its commutativity with the Laplace-Beltrami operators [13, 5], which in the reduced basis become diagonal matrices of the eigenvalues $\boldsymbol{\Delta}_1$ and $\boldsymbol{\Delta}_2$.

- As a last step, the estimated map $\mathbf{C}$ can be converted to a point-to-point map commonly by nearest neighbor search between the aligned spectral embeddings $\Phi_1\mathbf{C}^\top$ and $\Phi_2$, with possible post-refinement applied [46, 22, 42, 47].

To train the feature extractor network $\mathcal{F}_\Theta$, one defines a set of training shape pairs, and another set of shape pairs for testing. As shown in [5], the solution to Eq. (1) can be obtained in closed form within a neural network in a differentiable manner, and constitutes what is called "FMReg" in Fig. 2. Using this insight, during training time, the network aims at reducing a loss $L(\mathbf{C})$, defined on the output functional map $\mathbf{C}(\mathcal{F}_\Theta(\mathcal{S}_1), \mathcal{F}_\Theta(\mathcal{S}_2))$ estimated from the learned descriptors using the closed form solution of Eq. (1). Through backpropagation, the parameters $\Theta$ are then updated to make the network produce better features for the next pair of shapes.

We stress that in this pipeline, the size $k$ of the functional map is a critical non-learned hyperparameter, which can strongly affect matching results (as highlighted in Fig. 1 and in existing literature [9]).

## 4   Method

The main goal of our work is to robustly and adaptively estimate functional maps for shape pairs with diverse geometric properties, including both near-isometric and non-isometric transformations. In this section, we describe the technical details of our proposed non-rigid shape matching framework. We illustrate the whole pipeline in Fig. 2. Our framework has two main stages: multi-resolution functional map learning (Sec. 4.1) and spectral attention learning (Sec. 4.2).

### 4.1   Multi-resolution Functional Maps

In the first stage of our framework, given the input shapes $\mathcal{S}_1$ and $\mathcal{S}_2$, we follow the basic learning pipeline, as described in Sec. 3, to infer multi-resolution functional maps for extensively characterizing

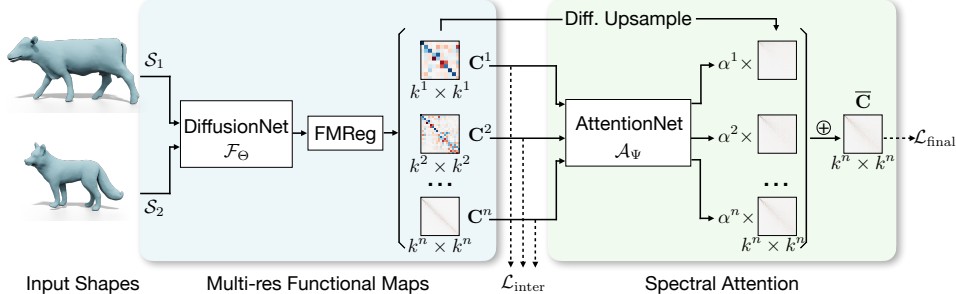

Figure 2: Illustration of our non-rigid shape matching pipeline based on multi-resolution functional maps with spectral attention. Our network takes as input a pair of shapes $\mathcal{S}_1$ and $\mathcal{S}_2$. First, we estimate a set of intermediate functional maps $\{\mathbf{C}^i\}_{i=1}^n$ with varying resolution, using DiffusionNet $\mathcal{F}_\Theta$ as the feature backbone and the differentiable FMReg module for the map computation. Next, we design a spectral attention network $\mathcal{A}_\Psi$ to predict a set of attention weights $\{\alpha^i\}_{i=1}^n$ for combining the functional maps at different resolutions, transformed by differentiable spectral upsampling, into the final map $\overline{\mathbf{C}}$. We impose losses $\mathcal{L}_{\text{inter}}$ and $\mathcal{L}_{\text{final}}$ during training.

shape matching in the spectral domain. As mentioned above, this differs from existing learning-based functional map works that typically compute *a single map* at a specific hand-picked resolution, which may not be universally applicable due to the rich diversity of shapes, thus putting limit on the generalization of networks across datasets.

To learn multi-resolution maps, we adopt a recent surface feature learning network DiffusionNet [14] as the backbone to learn the $d$ probe functions. DiffusionNet is shown to be robust to varying mesh sampling and widely applicable to non-rigid shape analysis tasks [14, 12]. Following the notation in Sec. 3, let $\mathcal{F}_\Theta$ denote this feature backbone. We feed the input shapes to the network in a Siamese manner and obtain $G_1 = \mathcal{F}_\Theta(\mathcal{S}_1)$ and $G_2 = \mathcal{F}_\Theta(\mathcal{S}_2)$. These learned probe functions are shared in the subsequent computation of multi-resolution maps.

To solve for a functional map with Eq. (1) in a differentiable manner, we employ the FMReg module proposed in [5]. With slight abuse of notation, let $\mathcal{C} = \{\mathbf{C}^i\}_{i=1}^n$ denote a series of $n$ estimated functional maps with varying resolution (Fig. 2-left), where the $i^{\text{th}}$ map $\mathbf{C}^i$ obtained via Eq. (1) is of size $k^i \times k^i$. Adjacent maps differ in their resolution by $\tau$ rows and $\tau$ columns, *i.e.*, $k^{i+1} = k^i + \tau$ for $\mathbf{C}^{i+1}$, meaning that $\tau$ more eigenfunctions are included in $\Phi_1$ and $\Phi_2$ for the map optimization.

**Acceleration** In the above approach, FMReg is repeatedly used to optimize Eq. (1) for each $\mathbf{C}^i \in \mathcal{C}$, incurring noticeable computation cost. To address this issue, we point out an acceleration scheme based on principal submatrices to significantly simplify the computation of $\mathcal{C}$. Specifically, we observe that for $i < n$, the principal $k^i \times k^i$ submatrix of $\mathbf{C}^n$ can be treated as an approximation of $\mathbf{C}^i$ computed in the standard way with Eq. (1). Accordingly, we can first employ FMReg once to compute the largest map $\mathbf{C}^n$ and then construct the multi-resolution functional maps $\mathcal{C}$ as a set of principal submatrices of $\mathbf{C}^n$, thus avoiding the optimization for $\{\mathbf{C}^i\}_{i=1}^{n-1}$. In practice, we find this acceleration scheme to work reasonably well, showing comparable matching performance with the above standard procedure (Sec. 5).

## 4.2 Spectral Attention

Given the estimated multi-resolution functional maps $\mathcal{C}$, there are two main questions: (1) how to prioritize the maps, and (2) how to address the resolution difference for combining the maps together. Thus in the second stage of our framework, instead of performing hard assignment, we propose to learn for each map $\mathbf{C}^i \in \mathcal{C}$ a soft attention weight $\alpha^i \in [0, 1]$, representing its contribution to the final map assembly. To enable the assembly, we also introduce a differentiable spectral upsampling module to align the map resolution within the network.

**Network** Intuitively, our spectral attention network takes as input, the set of functional maps $\mathcal{C}$ of different sizes, and predicts, for each functional map $\mathbf{C}^i$ in $\mathcal{C}$, a scalar weight $\alpha^i$, which represents the *confidence* associated with $\mathbf{C}^i$. To predict $\{\alpha^i\}_{i=1}^n$, the attention network needs to jointly assess the multi-resolution functional maps with their associated spectral information (Fig. 2-right). Directly

using the eigenfunctions as input signals to the network can be problematic due to the known issues of sign flipping and order changes [13]. In this work, we opt for spectral alignment residual as pointwise features and build the network upon a PointNet-based architecture [48].

Specifically, let $\Phi_1^i$ and $\Phi_2^i$ denote the spectral embeddings associated with $\mathbf{C}^i$, *i.e.*, matrices of the first $k^i$ eigenfunctions on $\mathcal{S}_1$ and $\mathcal{S}_2$, respectively, as defined in Sec. 3. For a point $q \in \mathcal{S}_2$, we define its spectral alignment residual $r_q^i$ as:

$$r_q^i = \min_{p \in \mathcal{S}_1} \delta_{qp}^i, \qquad \delta_{qp}^i = \|(\Phi_2^i[q])^\top - \mathbf{C}^i(\Phi_1^i[p])^\top\|_2, \tag{2}$$

where $\Phi_1^i[p]$ denotes the $p^{\text{th}}$ row of the matrix $\Phi_1^i$, similarly for $\Phi_2^i[q]$. We gather the residuals across $\mathcal{C}$ into a feature vector $\mathbf{r}_q = [..., r_q^i/\sqrt{k^i}, r_q^{i+1}/\sqrt{k^{i+1}}, ...]^\top$ for the point $q$, where the scaling factor $\sqrt{k^i}$ is to counteract the dimensionality difference across the spectral embeddings. The feature $\mathbf{r}_q$ can be interpreted as a form of *confidence* of each map $\mathbf{C}^i$ at the point $q$.

The vectors of spectral alignment residuals $\{\mathbf{r}_q\}_{q \in \mathcal{S}_2}$ collectively form an unordered set. We then feed this $n$-dimensional point set to our spectral attention network, denoted as $\mathcal{A}_\Psi$ with learnable parameters $\Psi$. We build $\mathcal{A}_\Psi$ with the classification architecture of PointNet [48] with feature transformations and global feature pooling. We forward the pooled global features through multi-layer perceptrons and the softmax function to obtain *spectral attention weights* $\alpha^i$ s.t. $\sum_{i=1}^n \alpha^i = 1$.

Note that the spectral alignment residuals, as input to our spectral attention network $\mathcal{A}_\Psi$, are computed using the Laplacian eigenbasis. Nevertheless, as we prove in the supplementary material, the residuals, and thus our network, are invariant under the changes of the Laplacian eigenbasis. In practice, we have also observed that the network $\mathcal{A}_\Psi$ is stable and trains well.

**Differentiable Spectral Upsampling**  Due to the dimensionality difference, the functional maps in $\mathcal{C}$ cannot be directly combined together with the learned spectral attention weights. Inspired by [42], we propose a differentiable spectral upsampling module to transform all the maps to the same spectral resolution. The main idea consists of two differentiable steps: (1) Convert the $k^i \times k^i$-size functional map $\mathbf{C}^i$ to a soft pointwise map; (2) Convert the pointwise map to a $k^n \times k^n$-size functional map. Specifically, we first compute a soft pointwise map, denoted as $\Pi^i \in \mathbb{R}^{n_2 \times n_1}$, from $\mathbf{C}^i$ as:

$$\Pi_{qp}^i = \frac{\exp(-\delta_{qp}^i/t)}{\sum_{p'} \exp(-\delta_{qp'}^i/t)}, \tag{3}$$

where $p, p' \in \mathcal{S}_1$ and $q \in \mathcal{S}_2$, and $t$ is a learnable temperature parameter. Next, to compute an upsampled map $\widehat{\mathbf{C}}^i$ of size $k^n \times k^n$, we project $\Pi^i$ onto the corresponding spectral basis:

$$\widehat{\mathbf{C}}^i = (\Phi_2^n)^\dagger \Pi^i \Phi_1^n, \tag{4}$$

where $\dagger$ denotes the Moore-Penrose inverse. After this differentiable upsampling, we obtain a set of functional maps $\widehat{\mathcal{C}} = \{\widehat{\mathbf{C}}^i\}_{i=1}^n$ that are all of size $k^n \times k^n$. Moreover, since they all represent maps between the same shape pair $\mathcal{S}_1$ and $\mathcal{S}_2$, they can be directly compared to each other and linearly combined.

**Map Assembly**  With the estimated spectral attention $\{\alpha^i\}_{i=1}^n$ and the upsampled functional maps $\widehat{\mathcal{C}}$, the last step of our framework is to assemble the intermediate maps together into a final coherent map output for shapes $\mathcal{S}_1$ and $\mathcal{S}_2$ (Fig. 2-rightmost). We denote the final map as $\overline{\mathbf{C}}$, which is computed by a simple linear combination as follows:

$$\overline{\mathbf{C}} = \sum_{i=1}^n \alpha^i \widehat{\mathbf{C}}^i. \tag{5}$$

### 4.3  Training

**Loss**  To train our network, we impose losses on both the intermediate multi-resolution functional maps $\mathcal{C} = \{\mathbf{C}^i\}_{i=1}^n$ and the final map output $\overline{\mathbf{C}}$, as shown in Fig. 2. Following prior works [5, 9], our network can be trained in a supervised or unsupervised way, and we present comprehensive results of both training paradigms in Sec. 5. In both the supervised and unsupervised settings, the loss $\mathcal{L}_{\text{inter}}$ for the intermediate maps $\mathcal{C}$ and the loss $\mathcal{L}_{\text{final}}$ for the final map $\overline{\mathbf{C}}$ are formulated as:

$$\mathcal{L}_{\text{inter}} = \frac{1}{n} \sum_{i=1}^n \left(\frac{k^n}{k^i}\right)^2 L(\mathbf{C}^i), \qquad \mathcal{L}_{\text{final}} = L(\overline{\mathbf{C}}), \tag{6}$$

where $L(\mathbf{C})$ is the penalty for a given map $\mathbf{C}$ depending on the training paradigm and described below. The map penalty in $\mathcal{L}_{\text{inter}}$ is multiplied by a scaling factor to counterbalance the spectral resolution difference. The overall training loss combines the two losses as $\mathcal{L}_{\text{total}} = \mathcal{L}_{\text{inter}} + \mathcal{L}_{\text{final}}$. We note that $\mathcal{L}_{\text{total}}$ remains unchanged regardless of whether the acceleration scheme (Sec. 4.1) is used.

For supervised training, we define the map penalty $L(\mathbf{C}) = \|\mathbf{C} - \mathbf{C}^{\text{gt}}\|^2$, where the Frobenius norm is used. The ground-truth functional map $\mathbf{C}^{\text{gt}}$ has the same spectral resolution as $\mathbf{C}$ and can be obtained by projecting the point-to-point map onto the reduced spectral basis. For unsupervised training, we follow [9] to define the map penalty as $L(\mathbf{C}) = \|\mathbf{C}^\top \mathbf{C} - \mathbf{I}\|^2$, which promotes orthogonality of the given map. Note that a Laplacian commutativity penalty [9] is not necessary because of the regularization imposed in Eq. (1).

**Implementation**   We implement our network with PyTorch [49]. The feature backbone consists of four DiffusionNet blocks [14] and produces $d = 256$ dimensional features as output. We use $\alpha = 10^{-3}$ in Eq. (1). During pre-processing, we perform a one-time eigendecomposition of the Laplacian on each shape and take the first 200 eigenfunctions, which are ordered by the corresponding eigenvalues and reused in all subsequent functional map computations. We compute $n = 20$ multi-resolution functional maps with sizes ranging from $10 \times 10$ to $200 \times 200$ with a step size $\tau = 10$. We set the batch size to 1 and use ADAM [50] for optimization. We use the WKS [29] descriptor as input to the network for all experiments, except for the SMAL dataset [6] in Sec. 5.2, where the 3D coordinates are used as input. This XYZ signal input is augmented with rotations around the up axis (here Y), since the shapes in SMAL are randomly rotated around this axis. This input signal can better address overfitting in the hard case of different animal species than WKS. More implementation details are provided in the supplementary material.

## 5   Experiments

In this section, we present extensive experimental results on a wide range of challenging non-rigid shape correspondence benchmarks to demonstrate the superior performance of our approach. We perform evaluations on widely adopted human shape matching datasets such as FAUST [2] and SCAPE [51] as well as a non-isometric dataset SHREC'19 [52]. We also evaluate on an animal shape matching dataset SMAL [6] to stress-test generalization. Moreover, we test on a recent humanoid shape dataset constructed from DeformingThings4D [53, 54] for both near-isometric and highly non-isometric correspondence. We use remeshed variants for these datasets, introduced in [22] and used in recent learning-based methods [5, 15, 11]. This ensures that the shapes do not share identical mesh connectivity.

### 5.1   Matching on FAUST, SCAPE, and SHREC'19

**Datasets**   Following [5, 11], the FAUST dataset, which has 100 human shapes, is split into 80 and 20 shapes for training and testing, respectively. For the SCAPE dataset consisting of 71 human shapes, the training and testing split is 51/20. The SHREC'19 dataset has 44 human shapes and is used only as a test set. Shape 40 in SHREC'19 is a partial shape and removed from the dataset, because the Laplacian basis used has global support [13, 26] and special architecture designs like [10] are required for partial shape matching, which is outside the scope of this work.

**Baselines**   We perform comparison with existing non-rigid shape matching works categorized as follows: (1) Axiomatic approaches including BCICP [22], ZoomOut [42], and Smooth Shells [55]; (2) Supervised learning approaches including FMNet [7], 3D-CODED [56], HSN [35], ACSCNN [57], TransMatch [58], and GeomFmaps [5]; (3) Unsupervised learning approaches including SURFM-Net [9], UnsupFMNet [8], NeuroMorph [39], DeepShells [11], and GeomFmaps [5].

3D-CODED and TransMatch take point clouds as input, while the other methods work on meshes. TransMatch is a recent transformer-based method typically requiring voluminous training data, thus in our experiments we initialized the training of TransMatch with its released network weights pretrained on the SURREAL dataset [59] consisting of 10,000 shapes.

GeomFmaps is a strong baseline closely related to our work. For a fair comparison, we also use DiffusionNet as the feature backbone in GeomFmaps, with the same input signal as ours, for improved performance [14]. The supervised and unsupervised training losses for GeomFmaps are the same as

Table 1: Mean geodesic error ($\times 100$) on **F**AUST, **S**CAPE, and **S**HREC'**19**

| Train | | **F** | | | **S** | | | **F + S** | | |
|---|---|---|---|---|---|---|---|---|---|---|
| Test | | **F** | **S** | **S19** | **F** | **S** | **S19** | **F** | **S** | **S19** |
| FMNet | sup | 11.0 | 30.0 | - | 33.0 | 30.0 | - | - | - | - |
| 3D-CODED | sup | 2.5 | 31.0 | - | 33.0 | 31.0 | - | - | - | - |
| HSN | sup | 3.3 | 25.4 | - | 16.7 | 3.5 | - | - | - | - |
| ACSCNN | sup | 2.7 | 8.4 | - | 6.0 | 3.2 | - | - | - | - |
| TransMatch | sup | 2.7 | 33.6 | 21.0 | 18.6 | 18.3 | 38.8 | 2.7 | 18.6 | 16.7 |
| TransMatch + Refine | sup | 1.7 | 30.4 | 14.5 | 15.5 | 12.0 | 37.5 | 1.6 | 11.7 | 10.9 |
| GeomFmaps | sup | 2.6 | 3.6 | 9.9 | 2.9 | 2.9 | 12.2 | 2.6 | 2.9 | 7.9 |
| Ours-Fast | sup | **1.3** | 2.9 | **7.1** | 1.8 | **1.8** | **11.7** | **1.3** | **1.8** | 7.1 |
| Ours | sup | 1.4 | **2.2** | 9.4 | **1.7** | **1.8** | 12.2 | **1.3** | **1.8** | **6.2** |
| BCICP | | 6.1 | - | - | - | 11. | - | - | - | - |
| ZoomOut | | 6.1 | - | - | - | 7.5 | - | - | - | - |
| SmoothShells | | 2.5 | - | - | - | 4.7 | - | - | - | - |
| SURFMNet | unsup | 15.0 | 32.0 | - | 32.0 | 12.0 | - | 33.0 | 29.0 | - |
| UnsupFMNet | unsup | 10.0 | 29.0 | - | 22.0 | 16.0 | - | 11.0 | 13.0 | - |
| NeuroMorph | unsup | 8.5 | 28.5 | 26.3 | 18.2 | 29.9 | 27.6 | 9.1 | 27.3 | 25.3 |
| DeepShells | unsup | **1.7** | 5.4 | 27.4 | 2.7 | 2.5 | 23.4 | **1.6** | 2.4 | 21.1 |
| GeomFmaps | unsup | 3.5 | 4.8 | 8.5 | 4.0 | 4.3 | 11.2 | 3.5 | 4.4 | 7.1 |
| Ours-Fast | unsup | 1.9 | **2.6** | **5.8** | **1.9** | **2.1** | **8.1** | 1.9 | **2.3** | 6.3 |
| Ours | unsup | 1.9 | **2.6** | 6.4 | 2.2 | 2.2 | 9.9 | 1.9 | **2.3** | **5.8** |

The **best** results are highlighted separately for supervised and unsupervised methods.

the map penalties $L$ defined in Sec. 4.3. GeomFmaps uses 30 eigenfunctions, as recommended in the original work [5].

**Results** Tab. 1 reports the matching performance in terms of mean geodesic error on unit-area shapes [60]. Ours-Fast refers to our method with the acceleration scheme (Sec. 4.1). We observe that our method outperforms all baselines in both the supervised and unsupervised settings. The slightly better correspondences obtained on FAUST by DeepShells [11] can be explained by the fact that they use refinement as a post-processing step, which our method *does not require*. Our method directly produces high-quality maps by choosing the best weighted combination of functional maps obtained from matching at different spectral resolutions. In particular, notice that how we improve upon the closest competitor, GeomFmaps [5], by enabling the network to optimize the spectral resolution.

Besides, the fast version of our method gives almost the same results as the complete approach, and even better results in some cases. We attribute this partly to the fact that Ours-Fast avoids solving many linear systems inside the network. While Ours is more principled, it also relies on solving multiple linear systems with differentiable matrix inversion for *each* intermediate functional map [5]. Numerically this can lead to more instabilities, especially at the early training stage when descriptors are not fully trained, which ultimately can lead to a drop in performance in certain cases. Nevertheless, the results indicate that the acceleration scheme used in Ours-Fast works well in practice.

## 5.2 Matching on SMAL

**Dataset** The SMAL dataset has 49 four-legged animal shapes of eight species. To stress-test the generalization of learning-based approaches, we use five species for training and three species for testing, resulting in a 29/20 split of the shapes. Thus no shapes similar to the testing data are seen during training, and the correspondence across species is highly non-isometric, presenting great challenges to learning-based approaches.

Table 2: Results ($\times 100$) on SMAL and DT4D-H

| | | SMAL | DT4D-H | |
|---|---|---|---|---|
| | | | intra-class | inter-class |
| GeomFmaps | sup | 8.4 | 2.1 | **4.1** |
| Ours-Fast | sup | 5.8 | 2.0 | 4.7 |
| Ours | sup | **5.3** | **1.8** | 4.6 |
| DeepShells | unsup | 29.3 | 3.4 | 31.1 |
| GeomFmaps | unsup | 7.6 | 3.3 | 22.6 |
| Ours-Fast | unsup | 5.8 | **1.2** | 14.6 |
| Ours | unsup | **5.4** | 1.7 | **11.6** |

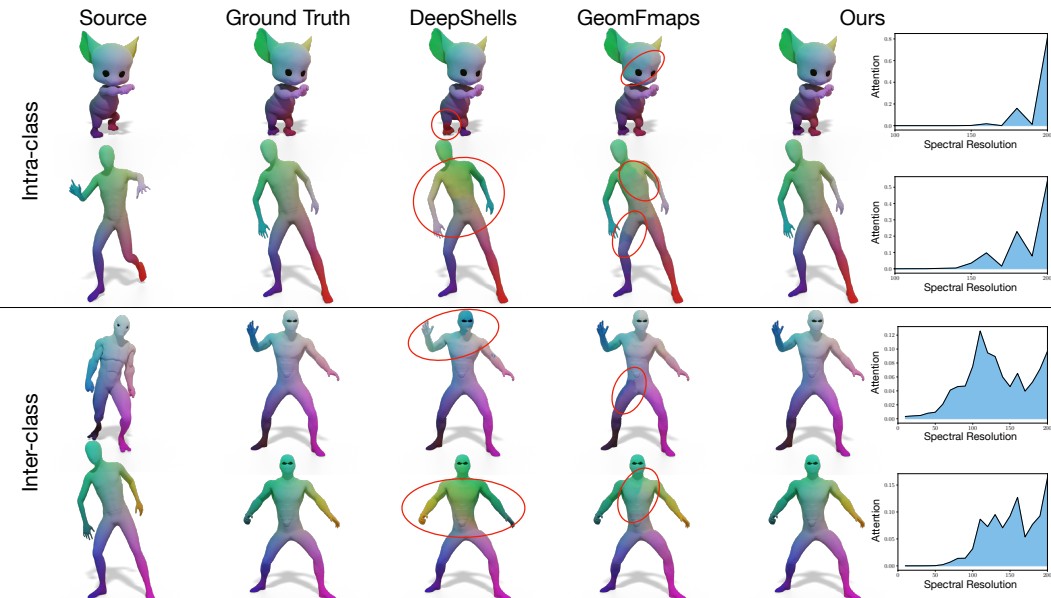

Figure 3: Correspondence visualization by color transfer for shapes from the DT4D-H dataset, where the methods are trained in the unsupervised setting. The rightmost plots are the learned spectral attention by our method.

**Results** Tab. 2-left reports the matching performance. Again, our method significantly improves upon its predecessor GeomFmaps in both supervised and unsupervised settings. Besides, DeepShells fails to predict accurate maps in this hard setting, due to its tendency to overfit to the training set, since they use SHOT input which is very triangulation-sensitive. Our method thus has strong generalization power, being able to predict correspondence between two animals whose species are never encountered during training.

## 5.3 Matching on DeformingThings4D

**Dataset** We also test on a recent non-rigid shape matching benchmark, introduced in [54]. This benchmark consists of shapes from a large-scale animation dataset DeformingThings4D [53] with dense ground truth correspondences. The inter-class correspondences were obtained by non-rigid ICP using manually selected landmarks [54]. We use nine classes of humanoid shapes for our evaluation. In total, there are 198/95 non-rigid, and often non-isometric, shapes for training/testing, significantly more than the previous datasets. We denote this humanoid shape benchmark as DT4D-H.

**Results** Tab. 2-right presents the performance for both intra and inter class matching. In the supervised setting, we obtain performance comparable to GeomFmaps. This may be due to local artefacts in the inter-class ground truth computed by non-rigid ICP across strongly non-isometric shape classes, with matching accuracy below 5 being saturated. Nevertheless, the improvement becomes remarkably noticeable in the unsupervised setting. DeepShells suffers from overfitting, as it is shown to succeed in intra-class matching but *not in inter-class matching*. Meanwhile, our approach is the only one to give reasonable overall results in this challenging unsupervised case.

**Visualization** In addition to the SMAL results in Fig. 1-(b), in Fig. 3 we show qualitative correspondence results on the DT4D-H dataset, across methods trained in the unsupervised setting. We observe that our method is able to produce high-quality correspondences for both intra and inter class shapes, compared to other methods.

In Fig. 3-right, we also plot the spectral attention (*i.e.*, $\{\alpha^i\}_{i=1}^n$ defined in Sec. 4.2) estimated by our network with the unsupervised training. The plots demonstrate that, for near-isometric (intra-class) shape pairs, the network puts most of the attention weights on functional maps of resolution in [150, 200]; while for more difficult non-isometric (inter-class) shape pairs, the network assigns higher weights to functional maps of resolution in [50, 150]. For the non-isometric case, this suggests that the smaller functional maps are more reliable and can be more accurately estimated than the larger ones. In contrast, for near-isometric shapes, higher-resolution maps can be estimated directly (and thus,

| Table 3: Ablation results ($\times 100$) | | |
|---|---|---|
| | Ours-Fast | Ours |
| Full model | 5.8 | 5.4 |
| w/o $\mathcal{L}_{\text{inter}}$ | 13.1 | 13.4 |
| w/o $\mathcal{A}_{\Psi}$ w/ uni. | 56.2 | 5.9 |
| w/o $\mathcal{A}_{\Psi}$ w/ avg. res. | 11.1 | 5.6 |
| $\tau = 5$ | 5.1 | 7.2 |
| $\tau = 20$ | 6.4 | 6.9 |

| Table 4: Backbone ablation ($\times 100$) | | |
|---|---|---|
| | DiffusionNet | KPConv |
| GeomFmaps | 8.4 | 9.4 |
| Ours-Fast | 5.8 | 6.8 |

lower-resolution ones as well, since those are given as principal submatrices of the larger functional maps). This highlights the importance of choosing the spectral resolution in a data-dependent manner.

### 5.4 Ablation Study

We perform ablation studies w.r.t. our network components and show the results in Tab. 3, where unsupervised training on SMAL is used. First, we train our network without the loss on intermediate multi-resolution functional maps and denote it as w/o $\mathcal{L}_{\text{inter}}$ in Tab. 3. We observe that using $\mathcal{L}_{\text{inter}}$ can greatly improve the matching performance owing to the better intermediate representations produced. Next, to validate the spectral attention network $\mathcal{A}_{\Psi}$, we remove it and compute spectral attention with two non-learning schemes. One option is to use a uniform weight (*i.e.*, $\alpha^i = 1/n$), denoted as w/o $\mathcal{A}_{\Psi}$ w/ uni. in Tab. 3. The other is to use mean spectral alignment residual (*i.e.*, softmax on $-\frac{1}{n_2 \sqrt{k^i}} \sum_{q \in \mathcal{S}_2} r_q^i$ for $\alpha^i$), denoted as w/o $\mathcal{A}_{\Psi}$ w/ avg. res. The results show that the matching performance degrades without the spectral attention network, in particular for Ours-Fast, which fails to converge with the uniform weight during training, strongly indicating the robustness brought by learned spectral attention. Moreover, we also test different values for the spectral step size $\tau$ (Sec. 4.1), and $\tau = 10$ is a balanced choice for both Ours-Fast and Ours.

To show the generality of our approach, we compare different feature backbones on SMAL where supervised training is used. Specifically, we replace DiffusionNet (Fig. 2) with KPConv [61], which is another advanced architecture working on point clouds. Tab. 4 shows that our method brings noticeable improvement across various backbones, indicating the wide generality of our approach.

## 6 Conclusion, Limitations, and Societal Impacts

To conclude, we introduced a robust non-rigid shape correspondence framework applicable in both supervised and unsupervised settings. We leverage a multi-resolution functional map representation and propose to learn spectral attention for a final coherent correspondence estimation, resulting in a powerful deep model that can adaptively accommodate near-isometric and non-isometric shape input. We demonstrate the improved matching performance of our model through extensive experiments on challenging non-rigid shape matching benchmarks.

One limitation of our approach is that, similarly to existing functional map works, we need to pre-compute spectral decomposition for each input shape. While the pre-computation is efficient for moderately sized shapes, down-sampling or remeshing may be needed for large-size input. Besides, we focus on full shape matching and assume input shapes represented as triangle meshes. It would be interesting to investigate an extension of our approach to partial shapes or point clouds as input.

Lastly, we do not see any immediate ethical issue with the proposed method, but note that considering the superior performance on human shape matching, unintended uses, such as surveillance, may be a potential negative societal impact of our work.

## Acknowledgements

The authors thank the anonymous reviewers for their valuable comments and suggestions. Parts of this work were supported by the ERC Starting Grant No. 758800 (EXPROTEA) and the ANR AI Chair AIGRETTE.

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
