# Learning Multi-resolution Functional Maps with Spectral Attention for Robust Shape Matching: Supplementary Material

**Lei Li**
LIX, École Polytechnique, IP Paris
lli@lix.polytechnique.fr

**Nicolas Donati**
LIX, École Polytechnique, IP Paris
nicolas.donati@polytechnique.edu

**Maks Ovsjanikov**
LIX, École Polytechnique, IP Paris
maks@lix.polytechnique.fr

In this supplementary material, we first provide a proof in Appendix A for the invariance of spectral alignment residuals under changes to the Laplacian eigenbasis. Next, we describe the architecture of our spectral attention network in Appendix B, and then introduce more implementation details in Appendix C. We give running time analysis in Appendix D. At last, we show more qualitative results in Appendix E.

## A    Basis Change in Spectral Alignment Residuals

In Section 4.2 of the main manuscript, we introduce spectral alignment residuals $\{r_q^i\}$, given by the multi-resolution functional maps $\{\mathbf{C}^i\}_{i=1}^n$, and use them as an input signal for our spectral attention network:

$$r_q^i = \min_{p \in \mathcal{S}_1} \delta_{qp}^i, \qquad \delta_{qp}^i = \|(\Phi_2^i[q])^\top - \mathbf{C}^i(\Phi_1^i[p])^\top\|_2. \tag{7}$$

Here we demonstrate that this residual input signal is in fact independant on the choice of source and target Laplacian eigenbasis $\Phi_1$ and $\Phi_2$, which is not trivial *a priori*.

Our first remark is that, due to sign flipping and order changes [1], two different Laplacian eigenbases may differ by a rotation matrix of size $k \times k$, which is a block-diagonal matrix with a rotation for each eigen space of the Laplacian up to size $k$. For eigen spaces of size 1, a rotation simply reduces to a potential sign flip, but it may be more general if the eigen space associated with a specific eigenvalue is of dimension 2 or more.

As a consequence, if we use another basis $\Psi_1$ and $\Psi_2$ on respectively the source and target shapes, we have the property that $\Psi_j^\dagger \Phi_j$ is such a block-diagonal matrix $\mathbf{R}_j$, with each block a rotation in the corresponding eigen space of the Laplacian on shape $j \in \{1, 2\}$.

**Theorem 1.** *Let $G_1$ and $G_2$ be (here fixed) per-point features on respectively source and target shapes, and $\mathbf{A}_1$ and $\mathbf{A}_2$ their respective projection in spectral space using $\Phi_1$ and $\Phi_2$ as the Laplacian eigenbasis. Let $\mathbf{C}(\Phi_1, \Phi_2)$ be the functional map solution to the minimisation problem:*

$$\mathbf{C} = \arg\min_{\mathbf{C}} \|\mathbf{C}\mathbf{A}_1 - \mathbf{A}_2\|^2 + \alpha\|\mathbf{C}\boldsymbol{\Delta}_1 - \boldsymbol{\Delta}_2\mathbf{C}\|^2, \tag{8}$$

*Then we have that a change in eigenbasis to $\Psi_1$ and $\Psi_2$ in that minimization problem results in a new functional map solution taking the following form:*

$$\mathbf{C}(\Psi_1, \Psi_2) = \mathbf{R}_2\mathbf{C}(\Phi_1, \Phi_2)\mathbf{R}_1^\top, \tag{9}$$

*with $\mathbf{R}_j = \Psi_j^\dagger \Phi_j$, for $j \in \{1, 2\}$.*

36th Conference on Neural Information Processing Systems (NeurIPS 2022).

*Proof.* It is proved in [2] that the solution to Eq. 8 is given in closed form by:

$$[\mathbf{C}^\top]_i = [(\mathbf{A}_1\mathbf{A}_1^\top + \alpha\mathbf{D}_i)^{-1}\mathbf{A}_1\mathbf{A}_2^\top]_i,$$

where $\mathbf{D}_i = \text{diag}(\lambda_s^1 - \lambda_i^2, s \in [1,k])^2$ ($\lambda^j$ being the eigenvalues of the Laplacian on shape $j \in \{1,2\}$), and $[X]_i$ denotes the $i^{\text{th}}$ column of the matrix $X$, with slight abuse of notation.

Firstly, let us denote by $\hat{\mathbf{A}}_j$ the features in the new eigenbasis $\Psi_j$. Then we have:

$$\hat{\mathbf{A}}_j = \Psi_j^\dagger G_j = \Psi_j^\dagger \Phi_j \mathbf{A}_j = \mathbf{R}_j \mathbf{A}_j$$

Then:

$$\hat{\mathbf{A}}_1\hat{\mathbf{A}}_1^\top + \alpha\mathbf{D}_i = \mathbf{R}_1(\mathbf{A}_1\mathbf{A}_1^\top + \alpha\mathbf{D}_i)\mathbf{R}_1^\top$$

And:

$$\hat{\mathbf{A}}_1\hat{\mathbf{A}}_2^\top = \mathbf{R}_1\mathbf{A}_1\mathbf{A}_2^\top\mathbf{R}_2^\top$$

Consequently:

$$(\hat{\mathbf{A}}_1\hat{\mathbf{A}}_1^\top + \alpha\mathbf{D}_i)^{-1}\hat{\mathbf{A}}_1\hat{\mathbf{A}}_2^\top = \mathbf{R}_1\Big((\mathbf{A}_1\mathbf{A}_1^\top + \alpha\mathbf{D}_i)^{-1}\mathbf{A}_1\mathbf{A}_2^\top\Big)\mathbf{R}_2^\top$$

**Lemma 1.** *Given the dependence of $\mathbf{D}_i$ on the eigenvalues of the second shape and the fact that the rotation $\mathbf{R}_2$ is block diagonal for eigenspaces (also of the second shape), we have:*

$$[\Big((\mathbf{A}_1\mathbf{A}_1^\top + \alpha\mathbf{D}_i)^{-1}\mathbf{A}_1\mathbf{A}_2^\top\Big)\mathbf{R}_2^\top]_i = [\mathbf{C}^\top\mathbf{R}_2^\top]_i$$

*Proof.* Indeed,

$$[\Big((\mathbf{A}_1\mathbf{A}_1^\top + \alpha\mathbf{D}_i)^{-1}\mathbf{A}_1\mathbf{A}_2^\top\Big)\mathbf{R}_2^\top]_{l,i} = \sum_{h\in[1,k]}[\Big((\mathbf{A}_1\mathbf{A}_1^\top + \alpha\mathbf{D}_i)^{-1}\mathbf{A}_1\mathbf{A}_2^\top\Big)]_{l,h}[\mathbf{R}_2^\top]_{h,i}$$

where $[X]_{l,i}$ denotes the term on the $l^{\text{th}}$ row, $i^{\text{th}}$ column of the matrix $X$.

Given that $[\mathbf{R}_2^\top]_{h,i} = 0$ when $\lambda_h^2 \neq \lambda_i^2$, because of the block-diagonal structure of $\mathbf{R}_2$, we can only keep the terms in the sum where $\lambda_h^2 = \lambda_i^2$. Then for these terms, we have $\mathbf{D}_i = \mathbf{D}_h$, because the dependence is precisely on the eigenvalues of the second shape.

Now, using that $[(\mathbf{A}_1\mathbf{A}_1^\top + \alpha\mathbf{D}_h)^{-1}\mathbf{A}_1\mathbf{A}_2^\top]_h = [\mathbf{C}^\top]_h$, we can individually rewrite the individual terms, to get:

$$[\Big((\mathbf{A}_1\mathbf{A}_1^\top + \alpha\mathbf{D}_i)^{-1}\mathbf{A}_1\mathbf{A}_2^\top\Big)\mathbf{R}_2^\top]_{l,i} = \sum_{h\in[1,k],\lambda_h^2=\lambda_i^2}[\mathbf{C}^\top]_{l,h}[\mathbf{R}_2^\top]_{h,i}$$

$$=[\mathbf{C}^\top\mathbf{R}_2^\top]_{l,i}$$

which concludes the lemma. □

Finally, if we denote $\hat{\mathbf{C}} = \mathbf{C}(\Psi_1, \Psi_2)$, we get:

$$[\mathbf{R}_1^\top\hat{\mathbf{C}}^\top]_i = \mathbf{R}_1^\top[\hat{\mathbf{C}}^\top]_i$$
$$= [(\mathbf{A}_1\mathbf{A}_1^\top + \alpha\mathbf{D}_i)^{-1}\mathbf{A}_1\mathbf{A}_2^\top\mathbf{R}_2^\top]_i$$
$$= [\mathbf{C}^\top\mathbf{R}_2^\top]_i \qquad\qquad \text{(because of Lemma 1)}$$

which concludes the proof. □

Using Theorem 1, we then have:

$$\delta_{qp}^i(\Psi_1, \Psi_2) = \|(\Psi_2^i[q])^\top - \mathbf{C}^i(\Psi_1, \Psi_2)(\Psi_1^i[p])^\top\|_2$$
$$= \|\mathbf{R}_2\big[(\Phi_2^i[q])^\top - \mathbf{C}^i(\Phi_1, \Phi_2)(\Phi_1^i[p])^\top\big]\|_2$$
$$= \delta_{qp}^i(\Phi_1, \Phi_2) \qquad\qquad \text{(rotations do not affect } \ell 2 \text{ norm)},$$

which proves that our spectral alignment residuals are indeed independant on the choice of basis.

# B  Spectral Attention Network

In Fig. 4, we illustrate the PointNet-based[3] architecture of our spectral attention network. Following the notation in Sec. 4.2, the input is the $n$-dimensional spectral alignment residuals $\{\mathbf{r}_q\}_{q \in \mathcal{S}_2}$, and the output is the predicted spectral attention $\{\alpha^i\}_{i=1}^n$. To extract global features in the network, we use `GlobalAvgPool`, which computes a weighted mean according to point areas for discretization invariance as follows:

$$g^l = \frac{\sum_q m_q f_q^l}{\sum_q m_q},\tag{10}$$

where $g^l \in \mathbb{R}$ denotes the $l^{\text{th}}$ dimension of the pooled global feature vector, $f_q^l \in \mathbb{R}$ denotes the $l^{\text{th}}$ dimension of the learned feature vector at the point $q$ from previous layers, and $m_q \in \mathbb{R}$ is the local area at the point $q$.

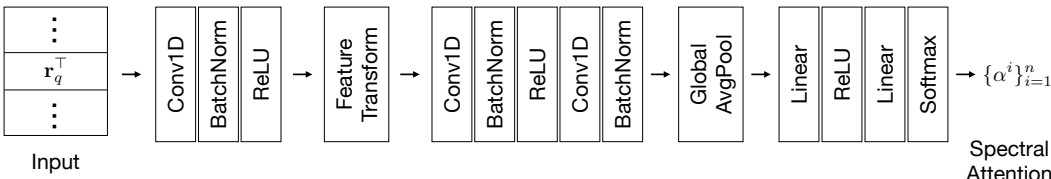

Figure 4: Architecture of our spectral attention network based on PointNet.

# C  Implementation

In this section, we provide more implementation details of our model to complement Sec 4.3.

To train on the FAUST, SCAPE, and DT4D-H datasets, we use the Wave Kernel Signature (WKS) [4] as input signal to the feature extractor backbone DiffusionNet. The dimensionality of WKS is 128. The initial learning rate is set to $10^{-4}$ and decayed by 0.1 at the half of training. The same settings are applied to the baseline GeomFmaps [2].

The SMAL dataset has eight species of four-legged animal shapes. The training data is composed of five species, including cow (8 shapes), dog (9), fox (4), lion (5), and wolf (3). The testing data is composed of the remaining three species, including cougar (4), hippo (6), and horse (10). Thus no shapes similar to the testing data are seen during training. To train on SMAL, we use the 3D coordinates as input signal to the network. We observed overfitting with WKS on SMAL for both GeomFmaps and our model, mainly due to the limited training data (29 shapes) and the challenging setting of no species overlap between the training and testing data. Thus we opt for the XYZ signal input augmented with random rotations around the up (or Y) axis. The initial learning rate is set to $10^{-3}$. The same settings are applied to the baseline GeomFmaps.

We use servers equipped with NVIDIA TITAN RTX and GeForce RTX 2080 Ti GPUs for network training.

# D  Running Time

We show the running time of our approach and the baseline GeomFmaps in Tab. 5. The statistics were collected on a server with Intel Xeon CPU @ 2.20GHz, 64GB RAM, and NVIDIA GeForce RTX 2080 Ti GPU. The computation of $n = 20$ multi-resolution functional maps with the FMReg layer [2] is the bottleneck of our approach (*i.e.*, in the *Functional Map* column of Tab. 5). However, we observe that using the acceleration scheme based on principal submatrices (*i.e.*, Ours-Fast) significantly reduces the running time. The spectral attention network and differentiable spectral upsampling have moderate computation cost. Nevertheless, we will investigate further optimizations on our model in future work.

Table 5: Running time (s) per shape pair averaged on SMAL

|  | Functional Map | AttentionNet | Diff. Upsample | Total |
|---|---|---|---|---|
| GeomFmaps | 0.053 | - | - | 0.053 |
| Ours-Fast | 0.221 | 0.013 | 0.039 | 0.273 |
| Ours | 1.303 | 0.013 | 0.039 | 1.355 |

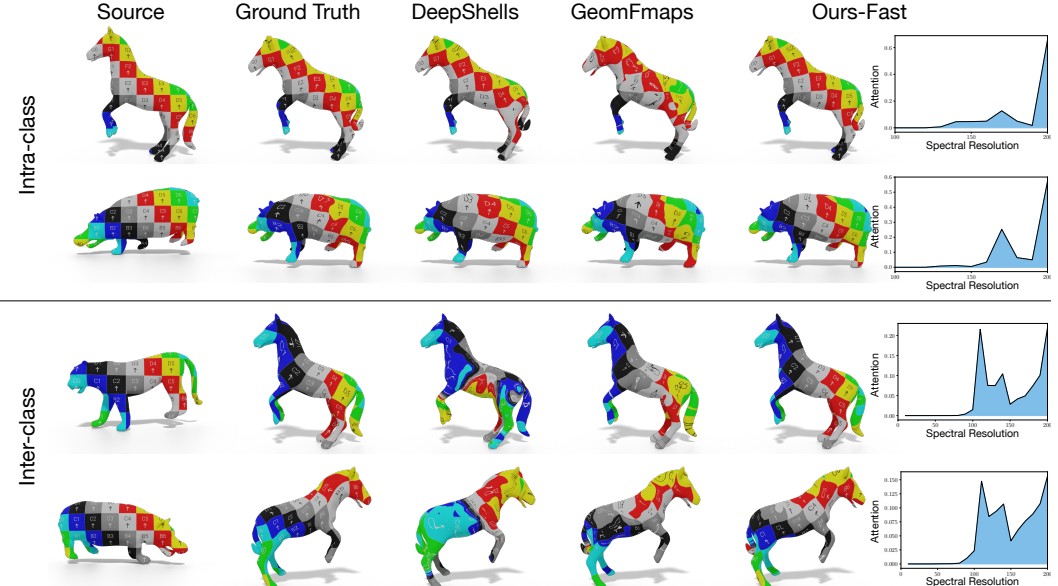

Figure 5: Correspondence visualization by texture transfer for shapes from the SMAL dataset, where the methods are trained in the unsupervised setting. Note that the species of the testing animal shapes are never seen during training.

## E  Qualitative Results

In Fig. 5, we present more qualitative results of non-rigid shape matching on the challenging SMAL dataset, where the compared methods are trained in the unsupervised setting. We reiterate that the evaluation on SMAL is a highly challenging stress-test for generalization, since there is no species overlap between the training and testing data, as mentioned in Appendix C. We observe that, compared to DeepShells [5] and GeomFmaps, our approach is able to produce more accurate correspondence for both near-isometric and non-isometric animal shapes, whose species are never seen during training, by learning to distribute attention weights across the spectral resolutions in an input data dependent manner.