# OpenReview forum: "Learning Multi-resolution Functional Maps with Spectral Attention for Robust Shape Matching"
_NeurIPS.cc/2022/Conference — NeurIPS 2022 Accept_

### Official Review · Reviewer_da7e · 2022-07-09

**Rating:** 6
**Confidence:** 4
**Soundness:** 3 good
**Presentation:** 3 good
**Contribution:** 2 fair

**Summary:**

The paper proposes a learning-based framework for non-rigid shape matching by building upon the functional map representation, and especially its learning-based variant GeomFmaps. The major contribution of the paper is learning the soft thresholding weights to combine functional maps obtained from matching at different spectral resolutions, which allows the network to optimize the spectral resolution. The paper illustrates the effectiveness of the combination on various benchmarks, on which it achieves state-of-the-art or comparable performance.

**Questions:**

1) line 47 -  "our framework consists **in** two novel.."  -- Did you mean "consists in" or "consists of"?
2) "Shape 40 in SHREC’19 is a partial shape and removed from the dataset, since it is outside the scope of this work" - it is not clear why partial shape is out of scope of this work, please explain (line 123 actually mentions that attention has been previously applied to partial shape matching)
3) Please explain why the fast method yields improved results in many cases (Tables 1 and 2).

**Limitations:**

To the best of my understanding there is no negative societal impact to the paper. The authors mention human surveillance, which does not  seem to be a direct application of the shape matching.
Otherwise the authors discuss general limitations -  computationally non-efficient computation of spectral decomposition for each input shape. (lines 360 - 363 "While the pre-computation is efficient for moderately sized shapes, down-sampling or remeshing may be needed for large-size input").

**Strengths And Weaknesses:**

Strengths:
Extensive experiments and state-of-the-art performance - the authors perform evaluations on widely adopted human shape matching datasets such as FAUST and SCAPE and a non-isometric dataset SHREC’19. Table 1 shows that on these datasets the paper outperforms the GeoFmaps (the method extended in this paper) and also outperforms all the other methods (presented), with a few exceptions.
The authors further evaluate on an animal shape matching dataset SMAL and show (Table 2) considerable improvement on GeomFmaps (e.g., in supervised setting: 5.3 vs. 8.4 mean geodesic error X 100) and DeepShells (unsupervised setting 5.3 vs. 29.3 ). Additional evaluation is provided on the challenging humanoid shape dataset from DeformingThings4D  for both near-isometric and
highly non-isometric correspondence. The authors show improved performance intra-class in supervised setting and intra and inter- class in unsupervised setting (Table 2). The authors also provide an ablation study Table 3, that shows the importance of learned spectral attention (linear combination weights of the functional maps).


Weaknesses:
1) Limited novelty of the idea:  lines (54 -  57) "Finally, to train our network, we propose to impose penalties on the intermediate multi-resolution functional maps as well as the final map. This is different from existing approaches [7, 9, 5, 10], to name a few, that penalize a single hand-picked spectral resolution" --Essentially the only novelty of this paper is learning of the soft-thresholding of the / best weighted combination of functional maps obtained from matching at different spectral resolutions. While the benefit to the performance is substantial, the idea is not essentially novel (it makes sense that a learned combination of weights would work better).
2) Extension of an existing work: The authors claim that their method is generic enough to directly benefit from other advances in deep functional map training, such as improved architectures or regularization. Note that this claim seems reasonable based on the architecture introduced in Figure 2 (the blue block can be probably replaced with a different architecture), however it is not supported by experiments. This makes their work seem like an extensions of one specific approach.
3) There is little deep dive into the results, for example the authors do not elaborate on why the more efficient technique named "Ours-FAST" sometimes outperforms "Ours", which is not intuitive.  The authors only say  lines 293 - 295: "Besides, the fast version of our method gives almost the same results as the complete approach, and  even better results in some cases. This underlines the fact that the approximation made in this version of our model works well in practice.". There is no insight as to why this happens.

---

> ### Author Response · Authors · 2022-08-01
> **Response to comments by Reviewer da7e**
>
> We would like to thank the reviewer for the constructive comments and acknowledging that
> 1. The proposed method learns the soft thresholding weights to combine functional maps obtained from matching at different spectral resolutions, which allows the network to optimize the spectral resolution.
> 2. Extensive experiments show the effectiveness of the combination on various benchmarks, on which it achieves state-of-the-art or comparable performance.
>
> Below we respond to the specific comments raised by the reviewer in detail.
>
> **Q1. Technical novelty.**
>
> We agree that intuitively a learned combination of weights should work better. Please note, however, that in order to enable this, we had to introduce multiple novel contributions such as, in-network spectral upsampling, which allows to compare and combine different spectral resolution functional maps. Furthermore, predicting the attention weights from the residuals of spectral alignment is also novel, and has not been considered before, to the best of our knowledge. All of these novel components are important for the performance and accuracy of our approach. Our technical contributions are appreciated by the reviewers **PHU8** and **6wTY**, who find our proposal sound and interesting.
>
> **Q2. Generality with other architectures.**
>
> We thank the reviewer for the suggestion. To demonstrate the generality of our approach, we performed a quick test replacing DiffusionNet (Fig. 2) with KPConv, which is another advanced feature extraction backbone, with supervised training. The table below shows that our method brings consistent improvement across different architectures in various testing settings, including the generalization stress-test on SHREC'19 and SMAL consisting of non-isometric shape pairs and unseen shape classes. Thus our approach is generic and *not* an extension of one specific approach.
>
> | Train on / Test on | FAUST+SCAPE / FAUST | FAUST+SCAPE / SCAPE | FAUST+SCAPE / SHREC'19 | SMAL / SMAL |
> |---|---|---|---|---|
> | GeomFmaps + DiffusionNet | 2.6 | 2.9 | 7.9 | 8.4 |
> | Ours-Fast + DiffusionNet | **1.3** | **1.8** | **7.1** | **5.8** |
> | GeomFmaps + KPConv | 2.7 | 6.6 | 13.3 | 9.4 |
> | Ours-Fast + KPConv | 1.4 | 5.5 | 8.1 | 6.8 |
>
> References:
>
> H. Thomas, et al. "*KPConv: Flexible and deformable convolution for point clouds.*" ICCV 2019.
>
> **Q3. Ours-Fast results.**
>
> While this behavior can indeed be counter-intuitive, we attribute it partly to the fact that Ours-FAST avoids solving many linear systems inside the network. While the full method (Ours) is more principled, it also relies on solving $k$ linear systems with differentiable matrix inversion for *each* intermediate functional map. Numerically this can lead to more instabilities, especially at the early training stage when descriptors are not fully trained. While Ours-FAST leverages principal submatrices to simplify the computation, we have the loss $L_{inter}$ to supervise all the intermediate functional maps at training time to ensure approximation accuracy (Sec. 4.3). Comparatively the full method can be more prone to numerical issues, which ultimately can lead to a drop in performance in certain cases.
>
> **Q4. Partial shapes.**
>
> Our work does not have an explicit component for overlap mask prediction between partial shapes, which needs to be considered in partial functional correspondence [24,10]. The Laplacian basis used in the functional map framework [13] has global support. When shapes are partially overlapped, a direct application of the functional map framework would not produce reasonable results, and special architecture designs like [10] are required for the overlap mask prediction. While this is possible, it is orthogonal to the scope of our work. We also remark that in [10], attention is learned in the feature domain. In contrast, our work learns attention in the spectral domain for functional maps, which has not been studied in existing literature.
>
> We hope to have addressed the reviewer's concerns in detail and would thus kindly ask the reviewer to reconsider their rating.

---

> > ### Comment · Reviewer_da7e · 2022-08-09
> > **Changing my decision to Weak Accept**
> >
> > I would like thank the authors for their hard work to address my comments (including performing additional experiments). After reading authors' response as well as other reviews + responses, I would like to change my decision to weak accept.
> > I would strongly recommend the authors to include all the clarifications and the additional experiments in the main paper (e.g., I have not seen KPConv experiment in the paper).

---

> > > ### Author Response · Authors · 2022-08-09
> > > **Response to follow-up comments by Reviewer da7e**
> > >
> > > We thank the reviewer for the positive feedback. We will, for sure, include all the clarifications and the additional experiments in our paper.

---

### Official Review · Reviewer_6wTY · 2022-07-11

**Rating:** 7
**Confidence:** 5
**Soundness:** 2 fair
**Presentation:** 3 good
**Contribution:** 2 fair

**Summary:**

The paper proposes a neural network architecture that exploits multi-resolution functional maps to predict a pointwise correspondence between two shapes. The architecture is composed of two blocks. The first block follows the idea of Deep Functional Maps (REF) and consists of multiple networks each predicting a functional map at a different resolution. The second block takes as input these functional maps and the residual (i.e. the eigenbases alignment error per each point of the two shapes) and computes an attention score with which to derive the final functional map as a weighted sum of all the functional maps. To make it possible to sum the maps at different resolutions, the authors propose to adopt a differentiable upsampling scheme inspired by ZoomOut. The method is validated on common 3D non-rigid datasets composed of humanoid and animal shapes.

**Questions:**

I’m a bit confused about the use of this attention scheme with the fast version of the network, where lower resolutions are derived as submatrices. In this case, it seems to me that the only difference between the different resolution FMs is given solely by the upsampling strategy. Are the upsampled matrices improving the final result or is most of the attention put on the larger FM? If for ZoomOut it is reasonable that, with few corresponding functions, solving for a smaller optimization problem and then upsampling can lead to better results, in this case, the method solves the least-square problem for the wider basis-function and then takes a submatrix. It is not obvious to me that this leads to the same result.

Also, following the previous point, is  L_{inter} for the fast version computed only for the wider FM? If so, why the huge difference in tab 3?

Minor:
- Maybe I missed it in the paper, but how are the point-wise correspondences computed?
- Description of deep functional map refers to a course. Isn’t there any reference to the original paper proposing this method?
- I’m also not sure of understanding the ablation of \tau. In tab 3 results with 5 and 20 are reported, while the full model should use 10, justifying this choice as a trade-off between speed and performance. Looking at the results I do not see this trend, in fact, both 5 and 20 are way worse than the Full model (with tau=10). Am I missing something?
- In the comparison with other methods, it is worth mentioning which kind of date each method works. Some of the methods take as input point clouds, while the proposed method requires triangular meshes (at least in the shown experiments). Also, even if not critical, it would be worth adding recent SOTA methods for shape registration with transformers (Trappolini et al., Shape registration in the time of transformers, NeurIPS 2021).


**Limitations:**

The paper clearly states the main limitations of the proposed approach, including the need for triangled meshes and the lack of experiments on partial ones. (eg. SHREC’16: Partial matching of deformable shapes, Matching deformable objects in clutter)

**Strengths And Weaknesses:**

The paper addresses a common problem in the design and application of Functional Map-based methods, which consists of the choice of suitable size of the truncated eigenbasis for the problem at hand. The proposed data-driven approach is for sure an interesting solution.

Unfortunately, an analysis of the obtained attention scores in different scenarios (e.g. near-isometric vs non-isometric) is missing and would have been a strong contribution to the paper. Indeed, as a motivation for this work, the authors suggest that few basis functions are prefered for highly non-isometric shapes while near-isometric shapes can benefit from the use of a wider eigenbasis that includes also higher frequencies. Even if this sounds like a reasonable premise, there isn’t any analysis supporting this claim.

As a last note, I found the paper generally well written, but some details on the method implementation are missing, or at least not perfectly clear.

---

> ### Author Response · Authors · 2022-08-01
> **Response to comments by Reviewer 6wTY**
>
> We would like to thank the reviewer for the constructive comments and acknowledging that
> 1. The paper proposes a neural network architecture that exploits multi-resolution functional maps to predict a pointwise correspondence between two shapes.
> 2. The proposed data-driven approach is for sure an interesting solution to address the choice of suitable size of the truncated eigenbasis, a common problem in functional map-based methods.
> 3. The method is validated on common 3D non-rigid datasets composed of humanoid and animal shapes.
>
> Below we respond to the specific comments raised by the reviewer in detail.
>
> **Q1. Attention score analysis.**
>
> Please note that we have provided analysis on the obtained attention scores in Sec. 5.3 (L330), in Fig. 3-rightmost of the paper, and in Fig. 5-rightmost of the supplementary material. The plots of the learned spectral attention show that, for near-isometric (intra-class) shape pairs, the network puts most of the weights on functional maps of resolution in [150, 200]; while for more difficult non-isometric (inter-class) shape pairs, the network assigns more weights to functional maps of resolution in [50, 150]. For the non-isometric case, this suggests that the smaller functional maps are more reliable and can be more accurately estimated than the larger ones. In contrast, for near-isometric shapes, higher-resolution maps can be estimated directly (and thus, lower-resolution ones as well, since those are given as principal submatrices of the larger functional maps). These results highlight the necessity of making the spectral resolution decision in a data-dependent manner, and our work presents the first study and an effective solution for this problem.
>
> **Q2. Fast version of the network.**
>
> The upsampled functional maps indeed improve the final result in the fast version of the network. As shown in Fig. 5-rightmost of the supplementary material, the fast version of the network learns to put attention weights on lower-resolution maps, especially in the inter-class setting. We remark that in Fig. 5, the text label "Ours" should be "*Ours-Fast*", and we will fix this typo in the final version. For the principal submatrices, we stress that, in addition to refinement by the upsampling strategy, they are also supervised by the $L_{inter}$ loss. Please refer to our response to **Q3** below.
>
> We performed an independent experiment w.r.t ZoomOut, according to the suggestion. Specifically, given a pair of shapes with extracted descriptors and the FMReg solver [5], we test the following two settings individually:
> 1. Compute an initial functional map of size 30X30 and then use ZoomOut to upsample it to a map of 200X200 as output.
> 2. Compute an initial functional map of size 200X200, take the principal submatrix of size 30X30, and then use ZoomOut to upsample it to a map of 200X200 as output.
>
> The table below shows that these two strategies lead to comparable results, further confirming the effectiveness of our principal submatrix-based idea.
>
> | id | FAUST | SCAPE | SHREC'19 | SMAL |
> |---|---|---|---|---|
> | 1. | 1.8 | 2.4 | 7.2 | 5.9 |
> | 2. | 1.8 | 2.4 | 6.9 | 6.1 |
>
> Finally, we stress that our method does *not* rely on the original ZoomOut method for post-refinement. Our differentiable spectral upsampling is an integral component of our network learning, and what is more important is that we apply it consistently in both training and testing stages.
>
> **Q3. $L_{inter}$ loss.**
>
> $L_{inter}$ is applied to *all* intermediate multi-resolution functional maps {$\mathbf{C}^{i}$} in both the fast and normal versions of the network, as shown in Eq. (6). The ablation study in Tab. 3 (Full model vs. w/o $L_{inter}$) shows that imposing $L_{inter}$ can help the network to produce better intermediate functional maps, which significantly improves the accuracy of the final output map. The results also show that $L_{inter}$ has similar effects on both Ours-Fast and Ours.
>
> **Q4. Implementation details.**
>
> We kindly refer the reviewer to the supplementary material (Sec. A & B) for more implementation details. We will be happy to provide clarifications on the implementation if requested. We will also make our code and data publicly available upon publication, to ensure full reproducibility of our work.

---

> > ### Author Response · Authors · 2022-08-01
> > **Response to comments by Reviewer 6wTY**
> >
> > **Q5. Minor issues.**
> >
> > **Q5-1.** Point-wise correspondence computation.
> >
> > To compute the point-wise correspondences from a functional map $\mathbf{C}$, we perform nearest neighbor search between the aligned spectral embeddings $\Phi_{1}\mathbf{C}^{\top}$ and $\Phi_{2}$, as mentioned in L153. We note that we did *not* perform any post-refinement for the results of our method (Tab. 1, 2, and 3).
> >
> > **Q5-2.** Reference of deep functional map.
> >
> > We will add references to the deep functional map works, including [7,8,9,5], in Sec. 3.
> >
> > **Q5-3.** The ablation of $\tau$.
> >
> > The reviewer's understanding of the ablation of $\tau$ is correct. We mean that $\tau=10$ is a balanced choice for both Ours-Fast and Ours to produce good performance. We will fix the misused words.
> >
> > **Q5-4.** Comparison with other methods.
> >
> > We will clarify the input data types of the compared methods. Specifically, 3D-CODED [53] works on point clouds but requires a template, and the other methods use meshes in the benchmarks.
> >
> > In the table below, we compare our work with the recent transformer-based method by Trappolini et al in the supervised setting. Our method has significantly better performance. In particular, this transformer-based method struggles on SCAPE and SHREC'19, which have more challenging human poses, and on SMAL, which stress-tests generalization to unseen shape classes. Besides, the small-scale training sets in these benchmarks may also pose challenges to this transformer-based method that requires voluminous training data. In contrast, our method works well with limited training data.
> >
> > | Train on / Test on | FAUST+SCAPE / FAUST | FAUST+SCAPE / SCAPE | FAUST+SCAPE / SHREC'19 | SMAL / SMAL |
> > |---|---|---|---|---|
> > | Trappolini et al. 2021 | 2.7 | 18.6 | 16.7 | 26.1 |
> > | Trappolini et al. 2021 + Refinement | 1.6 | 11.7 | 10.9 | 23.7 |
> > | Ours-Fast | **1.3** | **1.8** | 7.1 | 5.8 |
> > | Ours | **1.3** | **1.8** | **6.2** | **5.3** |

---

> > > ### Comment · Reviewer_6wTY · 2022-08-05
> > > **Few more clarifications**
> > >
> > > Thanks to the authors for clarifying my previous doubts.
> > > I just have a few more questions/consideration I would like to be addressed:
> > > 1 - I think the attention analysis is a crucial aspect of the analysis, I have to admit that I somehow missed it. Nevertheless, I would expand a bit its discussion on the main paper (and adjust the font size in fig 3).
> > > 2 - It is interesting (and somehow unexpected) your analysis of the ZoomOut upsampling. To better understand the analysis, it would be useful to show also the correspondence results directly using the 200x200 original FM.
> > > 3 - For the comparison with Trappolini et al., it would be fair to mention that you are putting yourself in the best conditions for your method (as you already noted, a small training set using the original architecture thought for a much larger training set). Further, Trappolini et al., similarly to 3D-CODED, work on PC and do not require connectivity as input.

---

> > > > ### Author Response · Authors · 2022-08-06
> > > > **Response to follow-up comments by Reviewer 6wTY**
> > > >
> > > > We would like to thank the reviewer for the positive feedback, and for the follow-up comments, which we address below.
> > > >
> > > > **Q6-1.** Attention analysis.
> > > >
> > > > We fully agree with the reviewer that the attention analysis is a crucial aspect of our work. We will, for sure, clarify and expand the discussion on this in our paper. We thank the reviewer for the suggestion on the writing.
> > > >
> > > > **Q6-2.** The independent experiment w.r.t ZoomOut.
> > > >
> > > > As suggested by the reviewer, we include the results of directly using the initial 200X200 functional map, labeled as id-3 in the table below (id-1 and id-2 are reported in a consistent way as in our previous response). We observe that id-3 has slightly better performance on FAUST and SCAPE consisting of near-isometric shapes, but it has significantly worse performance on SHREC'19 and SMAL, which are composed of non-isometric shapes never seen during descriptor training. We remark that the descriptors in this experiment are extracted by the same feature extractor as in Ours-Fast model trained in the supervised setting. Nevertheless, the result also echoes our attention analysis (**Q1**) and the utility of data-driven choice for spectral resolution.
> > > >
> > > > | id | FAUST | SCAPE | SHREC'19 | SMAL |
> > > > |---|---|---|---|---|
> > > > | 1. | 1.8 | 2.4 | 7.2 | 5.9 |
> > > > | 2. | 1.8 | 2.4 | 6.9 | 6.1 |
> > > > | 3. | 1.3 | 1.8 | 9.1 | 8.3 |
> > > >
> > > > **Q6-3.** Comparison with Trappolini et al.
> > > >
> > > > Thank you for the remark. We agree that comparing methods designed for different scenarios is not always straightforward. For fairness, we will provide in our paper, a fully detailed description on the training data size and input data type for the method by Trappolini et al.
> > > >
> > > > To clarify, for the comparison in the table of **Q5-4**, we obtained the results of the method by Trappolini et al. by initializing their network with their released trained weights and then fine-tuning on our used benchmarks. In our experiments, the results obtained by training their network from scratch on our used benchmarks (which are much smaller-scale than considered in that work) were not competitive. For the pre-trained weights, as mentioned in their paper, Trappolini et al. trained their network on the SURREAL dataset consisting of 10,000 shapes. Note that we do not pre-train our network on SURREAL, and we train from scratch on each dataset. We will make these points clear in our paper, and will be happy to include additional comparisons, if deemed appropriate by the reviewer.

---

### Official Review · Reviewer_PHU8 · 2022-07-11

**Rating:** 7
**Confidence:** 4
**Soundness:** 4 excellent
**Presentation:** 3 good
**Contribution:** 3 good

**Summary:**

The manuscript introduces shape matching framework based on multi-resolution functional maps. the idea is that the Laplace-Beltrami operators and their spectra are computed for shapes at various resolution, and for each resolution the corresponding functional map is computed. In order to make eigenfunctions and functional maps at various resolution comparable, a spectral upscaling scheme is adopted.
With he functional maps to hand, an attention mechanism based on mapped-point residual is adopted. The output of the attention network is a set of weight used to linearly combine the the maps.

**Questions:**

Does the approach assume uniform (possibly unequal) sampling of the shapes? I see no indication of the use of the area elements in the spectral decomposition and the multi-resolution approach would probably need to adapt to non-uniform sampling.



**Limitations:**

I believe that the negative societal impacts have been assessed correctly. As for Limitations, the performance issue with spectral decomposition is common to all spectral approaches, but they also incur other, arguably more severe limitations regarding their robustness to noise, extreme partiality, and topological deformation. That being said, the present proposal is affected by those limitations more than other spectral approaches because it needs to compute multiple spectral decomposition (one per scale) and then perform the upscaling since in the end it needs to work at the highest resolution.

**Strengths And Weaknesses:**

The addition of the multi-resolution aspect o functional maps is sound and interesting, and the attention mechanism provides a smart data-driven way to select the resolutions. However, what it lacks is locality: different scales can behave differently in different regions of the shape, especially in conjunction to non-uniform sampling.

I quite like the fact that the approach can work both in a supervised and unsupervised setting, but I believe that a supervised setting where the map is known is a bit unrealistic. Perhaps a setting where known corresponding functions over the two shapes are given would form a better use-case (incidentally, that was the original use-case for functional maps).

---

> ### Author Response · Authors · 2022-08-01
> **Response to comments by Reviewer PHU8**
>
> We would like to thank the reviewer for the constructive comments and acknowledging that
> 1. The multi-resolution functional map representation is sound and interesting, and the spectral attention mechanism provides a smart data-driven way to select the resolutions.
> 2. The approach can work both in the supervised and unsupervised settings.
>
> Below we respond to the specific comments raised by the reviewer in detail.
>
> **Q1. Shape sampling.**
>
> We clarify that our approach does *not* assume uniform sampling of the input shapes.
>
> First, we have taken into account the area elements of vertices of the input shape in the spectral decomposition of the Laplace-Beltrami operator (Sec. 3). Specifically, we follow the standard practice in existing literature to represent the Laplace-Beltrami operator as an $n \times n$ matrix $\mathbf{L} = \mathbf{S}^{-1} \mathbf{W}$. Here $\mathbf{S}$ is the diagonal matrix of lumped area elements for the $n$ vertices, and $\mathbf{W}$ is the cotangent weight matrix.
>
> Second, we use DiffusionNet [14] as the feature backbone, which also takes into account the area elements and is agnostic to varying mesh samplings.
>
> Therefore our approach is adaptive to non-uniform sampling by design. We will make these points clear in the paper.
>
> **Q2. Limitations of spectral decomposition.**
>
> We agree with the reviewer that the eigenfunctions of the Laplace-Beltrami operator, used in the classical functional map framework [13] and our work, are known to have certain limitations, such as lacking locality and being less robust to partiality and topological changes. In this way, we inherit the limitations of the works that we build upon, such as [5,7,9,8], etc.
>
> Please note, however, that the functional map framework in itself does not rely on a specific spectral basis. Thus regarding locality, our method can potentially be extended to other spectral bases that have more local support, such as the Compressed Manifold Modes proposed by T. Neumann et al., which are independent of mesh sampling and would be an interesting avenue for future work.
>
> We also clarify that our method does *not* need to compute multiple spectral decompositions. Indeed, for an input shape, we perform only a one-time eigendecomposition of the Laplacian matrix $\mathbf{L}$ defined above and take the first $k$ eigenfunctions, where $k$ corresponds to the *largest* functional map size (Sec. 4.1). These $k$ eigenfunctions are reused in the computation of multi-resolution functional maps (Sec. 4.1). Thus our method is *not more* affected by the above limitations than other spectral approaches.
>
> Nevertheless, we followed the evaluation protocols of existing non-rigid shape matching works [13,55,7,5,9], and our experiments (Tab. 1 & 2) show that our method robustly handles near-isometric and non-isometric shapes and generalizes well to unseen shape classes.
>
> References:
>
> T. Neumann, et al. "*Compressed manifold modes for mesh processing.*" CGF 2014.
>
> **Q3. Map supervision.**
>
> In the supervised setting, the ground-truth functional map can be obtained by projecting the ground-truth point-to-point map onto the reduced spectral basis. Note that a functional map of size $k \times k$ is fully determined by $k$ pointwise correspondences, i.e., a small number of precise landmarks on each shape. Such ground-truth labelings are readily available in the benchmarks tested in Sec. 5 and have been used in existing supervised methods [5,7]. The experiments (Sec. 5) show that our method can work well in the supervised setting in practice.
>
> Nevertheless, we agree that the unsupervised version of our method is particularly interesting as it can be applied in settings without any explicit supervision.

---

> > ### Comment · Reviewer_PHU8 · 2022-08-08
> > **Comments on authors' response**
> >
> > I want to thank the authors for their extensive replies.
> >
> > Q1: Good to know that you do take area elements into account, but it could be made clearer in the text (I did double check before posting and couldn't find any explicit reference and several places that lead to the misunderstanding)
> >
> > Q2: How do you compute only one decomposition and use it at all scales? do you compute it only at the highest resolution and use that for all scales assuming the implicit mapping given by the uspcaling? If so, again it is not very clear from the text.

---

> > > ### Author Response · Authors · 2022-08-08
> > > **Response to follow-up comments by Reviewer PHU8**
> > >
> > > We would like to thank the reviewer for the positive feedback, and for the follow-up comments, which we address below.
> > >
> > > **Q1**. We will, for sure, clarify the incorporation of area elements in our paper to avoid misunderstanding. We thank the reviewer for the careful check and the suggestion on the writing.
> > >
> > > **Q2**. We indeed compute the eigendecomposition of the Laplacian matrix $\mathbf{L}$ at the highest resolution (say, $k$). The resulting $k$ eigenfunctions are ordered by the corresponding eigenvalues. For a functional map of resolution $i \times i$, where $i < k$, we directly reuse the first $i$ eigenfunctions in the computation. We will make this point clear in the paper.

---

### Meta-Review · Area_Chair_vvZj · 2022-08-22

**Recommendation:** Accept
**Confidence:** Certain

**Metareview:**

All reviewers voted for acceptance of the paper. Reviewers acknowledge that the paper addresses an important problem: choosing the size of the truncated Eigenbasis for matching using functional maps. Also strong empirical performance on a number of datasets was noted. The rebuttal also addressed many points raised by reviewers and generally improved our impression of the paper. Overall this paper is a nice mix of theoretical contribution and practical performance. Therefore the paper is recommended for acceptance.
We ask the authors to incorporate the feedback given in the review and discussion phase.

**Award:**

No

---

### Decision · Program_Chairs · 2022-09-14

Accept